# Flight and Reproduction Variations of Rice Leaf Roller, *Cnaphalocrocis medinalis* in Response to Different Rearing Temperatures

**DOI:** 10.3390/insects12121083

**Published:** 2021-12-02

**Authors:** Weixiang Lv, Xingfu Jiang, Xiujie Chen, Yunxia Cheng, Jixing Xia, Lei Zhang

**Affiliations:** 1State Key Laboratory for Biology of Plant Diseases and Insect Pests, Institute of Plant Protection, Chinese Academy of Agricultural Sciences, Beijing 100193, China; lvwx@cwnu.edu.cn (W.L.); xfiang@ippcaas.cn (X.J.); xxjjchen@163.com (X.C.); yxcheng@ippcaas.cn (Y.C.); jixingxia@126.com (J.X.); 2Key Laboratory of Southwest China Wildlife Resources Conservation, China West Normal University, Nanchong 637002, China

**Keywords:** rearing temperature, development, reproduction, flight capacity, migration

## Abstract

**Simple Summary:**

Temperature directly affects the development, adult reproduction, and flight capacity in migratory insects. However, the adaptive strategies applied by some migratory insects to cope with stressful temperatures throughout their life cycles are not well understood. In this study, we evaluated the effects of rearing temperatures in the immature stage (from egg to pupae stage) on the immature development, adult reproduction flight ability, and migratory behavior of *Cnaphalocrocis medinalis*, one major facultative long-distance migratory pest feeding on rice. Our data suggest that immature *C. medinalis* that experienced different rearing temperatures had different developmental, reproductive, and migration patterns. *Cnaphalocrocis medinalis* reared under high temperatures had weaker reproductive capacity and stronger flight potentiality, which might be more likely to trigger the migration. However, those reared at low temperatures in the immature stage had an accelerated reproduction but relative weaker flight ability, which might weaken the migratory motivation of adults.

**Abstract:**

Understanding how species that follow different life-history strategies respond to stressful temperature can be essential for efficient treatments of agricultural pests. Here, we focused on how the development, reproduction, flight, and reproductive consequences of migration of *Cnaphalocrocis medinalis* were influenced by exposure to different rearing temperatures in the immature stage. We found that the immature rice leaf roller that were reared at low temperatures (18 and 22 °C) developed more slowly than the normal temperature 26 °C, while those reared at high temperatures (34 °C) grew faster. Female adults from low immature stage rearing temperatures showed stronger reproductive ability than those at 26 and 34 °C, such as the preoviposition period (POP) significantly decreased, while the total lifetime fecundity obviously increased. However, 34 °C did not significantly reduce the reproductive performances of females compared to 26 °C. On the contrary, one relative decreased tendency of flight capacity was found in the lower immature temperature treatments. Furthermore, flight is a costly strategy for reproduction output to compete for limited internal resources. In the lower temperature treatments, after d1-tethered flight treatment, negative reproductive consequences were found that flight significantly decreased the lifetime fecundity and mating frequency of females from low rearing temperatures in the immature stage compared to the controls (no tethered-flight). However, in the 26 and 34 °C treatments, the same flight treatment induced a positive influence on reproduction, which significantly reduced the POP and period of first oviposition (PFO). The results suggest that the experience of relative high temperatures in the immature stage is more likely to trigger the onset of migration, but lower temperatures in the immature stage may induce adults to have a greater resident propensity with stronger reproductive ability.

## 1. Introduction

Climate change influences insects at almost all levels, including distribution, phenology, and many other important biological and ecological processes [1,2]. Temperature is the major determinant factor among abiotic factors influencing the organismal performance and the response to biotic and abiotic stressors as well as the biological processes in insects [3,4,5,6]. Previous studies have widely reported variations of developmental rates, survival rates, fecundity, longevity, migratory behavior, and other life-history characteristics in response to rearing temperatures in Lepidoptera and other insect species [7,8,9,10,11,12]. Development and reproduction of insects are largely dependent on environmental temperatures, with optimal environmental temperatures having a positive effect. By contrast, high or low temperatures negatively impact reproduction and population growth [13,14,15]. Jiang et al. [16] reported that the development, reproduction, and longevity of *Mythimna separata* (Walker) (Lepidoptera: Noctuidae) exposed to high temperatures (above 30 °C) were restrained, as demonstrated in *Sitophilus granaries* (L.) (Coleoptera: Curculionidae) and other insect species [17,18]. Low temperatures reduced the development and fecundity in *Autographa gamma* (L.) (Lepidoptera: Noctuidae) [19] and delayed the pre-calling period in *M. separata* [20]. To ensure survival and reproduction, insects change their development and physiology under extreme conditions [21]. Ma et al. [1] revealed that the trade-off between reproductive traits and response to environmental impacts was an important part of the life history strategy in insects, which impaired the damage of extreme temperatures on the insects.

Seasonal migrations like many bird species occur on a large scale and frequently in many migratory insects [22]. Long-distance migration behavior has been an important adaptive life history strategy for population maintenance and evolution of migratory insects [23,24], which allows migratory species to leave deteriorating habitats and seek for new breeding habitats [25,26]. Energetic investments in migration and flight organs are costly for limited internal resources, which may reduce energy investment in reproduction at the result of producing fewer offspring [27,28]. Resource trade-off between reproduction and flight-associated characters during migration is common in most insects, such as *Aphis glycines* (Matsumura) (Hemiptera: Aphididae), *Spodoptera exempta* (Walker) (Lepidoptera: Noctuidae), and crickets [29,30,31], and this phenomenon is termed as the oogenesis-flight syndrome [32]. Zhang et al. [28] reported that migration-mediated reproductive costs were not absolutely negative but instead depended on the age, duration, and frequency of the moth’s initial flight. However, the trade-off between migration and reproduction under different rearing temperatures remains unknown in most long-distance migrants.

The rice leaf roller, *Cnaphalocrocis medinalis* (L.) (Lepidoptera: Pyralidae), is one of the predominant herbivorous pests in rice ecosystems, which has caused serious economic losses in Asia, Australia, and East Africa [33,34]. In China, it has been confirmed that *C. medinalis* cannot overwinter in the north area of 30° N [35]. It is most likely that migration in *C. medinalis* is an important adaption to seasonal and regional variations, similarly to *M. separata* [36,37]. Most studies have been conducted on the long-term monitoring and seasonal migratory mechanism of *C. medinalis* adults [34,38,39]. However, it remains unknown how the larval rearing temperature affects adult migration and reproduction of *C. medinalis*. Reproductive traits are sensitive to thermal variations, but the effect of stressful temperatures in early life stages on adult performance is still a matter of debate in insects [1]. Potter et al. [40] stated that early life stress had no effect on adult performance based on the adaptive decoupling hypothesis and life cycle modularity hypothesis. However, recent studies have found that stressful high temperatures in the immature stage (including egg, larval, or pupal stages) can decrease the fecundity of adults through carry-over effects [41,42] or even reduce offspring survival via trans-generational effects [43]. This raises an important question about whether the rearing temperature in the early life stage affects subsequent adult reproductive and flight performances. If yes, how do high or low temperatures in the immature stage affect the reproductive and migratory behavior of *C. medinalis*?

To gain insight into these questions above, we investigated the larval development, adult reproduction, flight capacity and reproductive consequences of migration of *C. medinalis* across the range of four constant immature stage temperatures under laboratory conditions. These results revealed that *C. medinalis* larvae and adults could deal with ambient temperature stress to ensure survival and reproduction. This study expands our knowledge on the population adaptation strategy of *C. medinalis* within complex ambient environments.

## 2. Materials and Methods

### 2.1. Insect Rearing

Rice leaf roller pupae were collected from rice fields in Changde, Hunan Province, China. The population was then reared under laboratory conditions (26 ± 1 °C, 85 ± 5% relative humidity (RH), and a photoperiod of 16L: 8D) to obtain eggs. Larvae in this experiment were reared with fresh corn seedlings 7–9 cm in height in the glass bottles (9 cm × 13 cm, diameter × height) until pupation. After pupation, the pupae were transferred from leaves in glass bottles with moist cotton wool on the bottom to optimize conditions. After emergence, each pair of female and male moths was immediately transferred into new cylindrical plastic cages (10 cm × 20 cm, diameter × height) and fed with fresh 10% honey solution (*v*/*v*) daily until the death of the adults [28].

### 2.2. Temperature Treatments

The upper and lower threshold temperatures for total development of *C. medinalis* were determined to be 36.4 and 11.2 °C [44]. To investigate the influences of rearing temperature on the immature development (from egg to pupae stage), subsequent adult reproduction, and flight performance, four temperature treatments (18, 22, 26, and 34 °C) were selected in this experiment. The control temperature was 26 °C, the optimal temperature for studying *C. medinalis* reproduction and flight behavior [24,35,36]. The eggs and subsequent hatched larvae were reared at the four constant rearing temperatures under a photoperiod of 16L: 8D with 85 ± 5% humidity until the eclosion of moths. Larvae were reared in culture plates (1 larval/well) and fed with maize seedlings at 85 ± 5% relative humidity (RH) with a 16:8 h light: dark.

### 2.3. Larval Development of C. medinalis

For each temperature treatment, eggs laid at the same time were gently collected and individually placed in every well of 6-well plates to observe hatching every day. Subsequently, all the hatched larvae were reared on fresh corn leaves until pupation, and the instar stage and number of surviving larvae were examined and recorded daily. The egg-larval period included the duration from egg hatching to pupation, and the pupal period was the duration between pupation and adult emergence. Subsequently, all hatched larvae were reared on fresh corn leaves until pupation, and the instar stage and number of surviving larvae were examined and recorded daily. Each temperature treatment had 3 replicates to determine the number of larvae and pupal mortality rate. More than 50 larvae or pupae were examined for each replicate. Similarly, pupation and eclosion rates were calculated based on the number of pupae and emerging adults. The period of synchronized eclosion was defined as the number of days between a pupa’s period of eclosion and the minimal period of eclosion among all pupae in the same treatment. It was used to measure the synchronization of adult emergence. The lower value of the period of eclosion represented the synchronized eclosion. The sex ratio (proportion of females) was used to calculate the proportion of female/total progeny [45]. The sample size of surviving larvae, pupae, and emerging adults from 18, 22, 26, and 34 °C treatments were 153, 356, 443, and 318; 114, 286, 340, and 244; and 110, 280, 291, and 236, respectively. Larvae or pupae that escaped before adult emergence were excluded from data analysis.

### 2.4. Reproductive Parameter Determination

This experiment was designed to evaluate the influence of larval rearing temperatures on adult reproduction. After emergence, all females and males from four rearing temperatures during the immature stages were immediately paired (one male and one female) at each temperature treatment and fed into cylindrical plastic cages (10 cm × 20 cm, diameter × height) at 26 ± 1 °C, 85–95% RH, with a photoperiod of 14L:10D. The oviposition of the adults was observed and recorded daily until death. The reproductive parameters were measured based on previous studies [46,47], which were used to evaluate the variations in response to different temperatures. The pre-oviposition period (POP) was an important parameter to describe the duration from new emergence to the first oviposition of adults. The period of first oviposition (PFO) was used to compute and evaluate the synchronization of first oviposition, which was defined as the period from a female’s POP to the minimal POP in a treatment. The lower PFO value showed the more synchronized initial oviposition [48]. After female death, the number of spermatophores was observed, and the mating frequency and mating percentage were calculated by the presence and number of spermatophores. The mating success was confirmed based on the presence of a spermatophore in the spermathecal. Females with spermatophores copulated successfully. Those females without spermatophores indicated failure of copulation [28]. Data were excluded from the analyses if either male or female escaped during the experiment. The experiments were terminated when all the males and females died. Sample sizes for each temperature treatment were 22, 42, 35, and 24 pairs from low to high rearing temperatures, respectively.

### 2.5. Flight Treatment and Capacity Assessments

The measurement of flight performance was conducted with a 48-channel flight mill system, which automatically recorded the flight parameters (flight duration, flight distance, and average flight velocity) [28,49].

All emerged moths including one-day-old females and males were anesthetized with ether, and the scales on the junction between the metathorax and abdomen were brushed away. A short hollow-plastic tether was glued to the dorsal surface of the metathorax using 502 super adhesives (Beijing Chemical Co, Beijing, China) and the adults were provided with 10% honey solution (*v*/*v*) before tethered flight. All tethered moths were tested for 6-h flight (23:00–05:00) on flight mills under dark conditions at 26 ± 1 °C and 80–85% RH [28,39,45]. The number of replicates under four rearing temperature treatments (18, 22, 26, and 34 °C) were 14, 28, 19, and 26 females, and 14, 30, 28, and 30 males, respectively. After the tethered flight test, females and males were immediately paired and fed in cylindrical plastic cages under laboratory conditions. The reproductive parameters mentioned above were observed and recorded daily until death. Moths that did not undergo tethered flight served as the control. Sample sizes for each temperature treatment were 22, 42, 35, and 24 females from low to high rearing temperatures, respectively.

### 2.6. Data Analysis

The larval period, pupal period and period of synchronized eclosion, flight capacity, and reproductive parameters of adults were analyzed using one-way analysis of variance (ANOVA) followed by Tukey’s honestly significant difference (HSD) test (*p* < 0.05) or independent samples *t*-tests (*p* < 0.05). Differences in all larval development parameters (except larval period, pupal period, and period of synchronized eclosion) among treatments were compared by the chi-square test. To compare the impact of flight treatment under different immature stage temperatures (18, 22, 26, and 34 °C) on adult reproduction, independent *t*-tests (ANOVA) were also used. All statistical analyses were performed with SPSS software (version 18.0; SPSS, Chicago, IL, USA).

## 3. Results

### 3.1. Development of C. medinalis Exposed to Different Rearing Temperatures in the Immature Stage

Rearing temperatures had a significant influence on *C. medinalis* from the egg to larval period (*F*_3,1266_ = 3768.66, *p* < 0.01) and pupal period (*F*_3,980_ = 847.94, *p* < 0.01). Low temperatures (18 and 22 °C) significantly extended the larval period and pupal period compared to the treatments of 26 and high temperature (34 °C), significantly delaying larval and pupal development under these temperatures. There was no significant difference in the larval period between the 34 and 26 °C treatments, but the pupal period of the 34 °C treatment was significantly shorter than that of the 26 °C treatment (Table 1).

Temperatures below or above 26 °C did not significantly increase the mortality rate of larvae and pupae (larval mortality rate: *χ*^2^ = 4.74, df = 3, *p* = 0.19; pupal mortality rate: *χ*^2^ = 3.36, df = 3, *p* = 0.34), nor reduce the pupation rate (*χ*^2^ = 0.85, df = 3, *p* = 0.84), eclosion rate (*χ*^2^ = 2.08, df = 3, *p* = 0.56), or sex ratio (*χ*^2^ = 0.85, df = 3, *p* = 0.84, Table 1). However, *C. medinalis* pupae exposed to low temperatures (18 and 22 °C) had a significantly higher period of synchronized eclosion than those at 26 and 34 °C (*F*_3,980_ = 76.36, *p* < 0.01, Table 1). There were no significant differences in the period of synchronized eclosion between the 34 and 26 °C treatments (Table 1).

### 3.2. Reproductive Performance of C. medinalis Adults from Different Rearing Temperatures in the Immature Stage (the Control Moths without Tethered Flight)

The POP of females was significantly affected by rearing temperature (*F*_3,119_ = 7.05, *p* < 0.01). Females at 18 and 22 °C had significantly shorter POPs than the moths at 26 and 34 °C, but there were no significant differences between the 34 and 26 °C treatments (Table 2). Rearing temperature had no significant effect on PFO (*F*_3,119_ = 1.22, *p* = 0.30, Table 2). However, the lifetime fecundity of females was also significantly affected by rearing temperature (*F*_3,119_ = 8.12, *p* < 0.01). Females at 18 and 22 °C laid more eggs than the moths at 26 and 34 °C, but no significant differences were found between the 34 and 26 °C treatments (Table 2).

Rearing temperature also significantly affected the mating frequency of females (*F*_3,119_ = 4.77, *p* < 0.01). Females at 18 °C had the greatest mating frequency, which was higher than those in the 26 and 34 °C treatments (Table 2). There were significant differences among female longevities for different rearing temperatures (*F*_3,119_ = 7.23, *p* < 0.01). Female longevity at 34 °C was significantly reduced compared to those at 18, 22, and 26 °C, but no significant differences were observed between the two low temperatures (18 and 22 °C) and 26 °C treatments (Table 2). Likewise, the oviposition period of moths at 34 °C was significantly lower than those in the 18 and 22 °C treatments, whereas it did not differ from the 26 °C treatment (*F*_3,119_ = 4.44, *p* < 0.01, Table 2). However, there was no significant differences in mating percentage (*F*_3,119_ = 1.85, *p* = 0.14, Table 2).

### 3.3. Flight Capacity of C. medinalis Emergenced from Different Rearing Temperatures in the Immature Stage

Rearing temperature had no significant effects on the flight capacity of female *C. medinalis*, but it significantly affected the flight capacity of males. Females at 26 °C had the longest flight duration and flight distance, which were higher than the other temperature treatments (flight duration: *F*_3,93_ = 2.05, *p =* 0.11; flight distance: *F*_3,93_ = 1.54, *p* = 0.21, Table 3). However, 34 °C significantly decreased the flight duration and flight distance of males compared to the moths exposed to 26 °C (flight duration: *F*_3,97_ = 4.16, *p* < 0.01; flight distance: *F*_3,97_ = 3.17, *p* = 0.05, Table 3). There were no significant differences in the average velocity of females (*F*_3,93_ = 0.51, *p* = 0.68) and males (*F*_3,97_ = 0.51, *p* = 0.68, Table 3).

Comparisons of the flight capacity between females and males showed that females exposed to 18 °C had a lower flight duration, flight distance, and average velocity than males; by contrast, these data were higher for males reared at higher temperatures. A significant decrease for flight duration was observed among the moths especially from the 34 °C treatment (*t*_53_ = 2.51, *p* = 0.02, Table 3).

### 3.4. Influence of Flight on Reproduction Treated by Different Immature Stage Rearing Temperatures

Flight significantly reduced the POPs of females treated with 26 and 34 °C compared to the controls at each larval rearing temperature (26 °C: *t*_56_ = −2.06, *p* = 0.04; 34 °C: *t*_45_ = −3.49, *p* < 0.01), which resulted in their early oviposition. However, flying females at 18 °C showed a significant delay in spawning (*t*_44_ = 2.98, *p* < 0.01, Figure 1A, Appendix A). The PFOs of moths at 34 °C were also significantly decreased by the flight compared to the controls at the same temperature (*t*_45_ = −3.49, *p* = 0.01), indicating that the synchronized oviposition of flying females exposed to high temperatures was better than the nonflying moths. Furthermore, flying females at 18 and 22 °C had higher PFOs than the controls, but the differences were not significant (Figure 1B, Appendix A). Flight also significantly decreased the lifetime fecundity (18 °C: *t*_44_ = −3.73, *p* < 0.01; 22 °C: *t*_65_ = −4.94, *p* < 0.01, Figure 1C, Appendix A) and mating frequency (18 °C: *t*_44_ = −3.25, *p* < 0.01; 22 °C: *t*_65_ = −2.38, *p* = 0.02, Figure 1D, Appendix A) of females exposed to 18 and 22 °C than the controls. Similarly, flying females at 22 and 26 °C had significantly shorter longevities than the control moths (22 °C: *t*_65_ = −3.41, *p* < 0.01; 26 °C: *t*_56_ = −2.99, *p* < 0.01, Figure 2A, Appendix A). Compared to the control moths, the oviposition period of females at 22 °C was significantly shortened by the flight (*t*_56_ = −3.85, *p* < 0.01, Figure 2B, Appendix A). Similarly, flight significantly decreased the mating percentage of females at 18 and 22 °C compared to the controls (18 °C: *t*_44_ = −2.464, *p* = 0.02; 22 °C: *t*_65_ = −1.98, *p* = 0.05, Figure 2C, Appendix A).

## 4. Discussion

In this study, we found that there was a relative decreased tendency of larval development and flight capacity of *C. medinalis* reared at below or above the optimal temperature in the immature stage (including egg, larval, or pupal stages), which was consistent with previous findings [12,50]. Low rearing temperatures (18 and 22 °C) significantly delayed the larval development but accelerated adult reproduction compared to the 26 °C and high rearing temperature (34 °C) treatments. However, 34 °C significantly accelerated the maturation development compared to 26 °C, but it did not significantly delay adult oviposition and decrease egg production. The results suggested that female adults from low temperatures in the immature stage showed stronger reproductive ability. Migration, as an adaptive life history strategy, must provide reproductive benefits over the remaining sedentary or at least on average for population maintenance and evolution [28,37,46]. The reproductive consequences of d1-tethered flight showed that flight also significantly affected the reproduction of adults that experienced different temperature treatments in the immature stage. Flight favored the migration of *C. medinalis* adults from high rearing temperatures (34 °C) in the immature stage, via shortening the preoviposition period and increasing the synchronized oviposition. By contrast, flight significantly reduced the lifetime fecundity and mating frequency of moths from low rearing temperatures in the immature stage (18 and 22 °C), thereby reducing the motivation of migration. These results may reflect adaptive ecological strategies to different environmental temperatures for *C. medinalis*.

The development of *C. medinalis* differed significantly under different rearing temperatures in the immature stage. Temperatures below and above 26 °C caused higher larval and pupal mortalities, and lower pupation and eclosion rates. By contrast, *C. medinalis* larvae and pupae developed well at the normal temperature of 26 °C, with the lowest mortality and the highest pupation rate and eclosion rate. Thus, these results were consistent with previous reports that stressful low and high temperatures negatively influenced the larval growth and development [51,52], such as for *Spodoptera exigua* (Hübner) (Lepidoptera: Noctuidae) [53], *M. separata* (Walker) [18], and other lepidopteran insect species. As expected, low temperatures resulted in a slower growth rate and longer development time in insects [54], while high temperatures accelerated the development of the egg, larval, and pupal period [49]. Our results also found that low rearing temperatures (18 and 22 °C) significantly slowed down the larval development and caused poor synchronized eclosion of *C. medinalis*, with significant increases in the larval and pupal period as well as the lengthening of the period of synchronized eclosion compared to 26 °C. However, high rearing temperatures (34 °C) reduced the pupal period but accelerated maturation processes. These results were also consistent with the results of Chintalapati et al. [44] for *C. medinalis* that showed that the developmental periods of different stages were reduced with increases in temperature from 18 to 34 °C.

Our results also showed that high rearing temperatures (34 °C) in the immature stage negatively affected the reproduction and longevity of adults. However, low rearing temperatures positively stimulated adult reproduction, which resulted in decreased POP and increased egg production of females. These findings suggested that rearing temperature in the immature stage could influence the reproductive performance of *C. medinalis* adults. *Cnaphalocrocis medinalis* is one of the important migrant species, which differentiated into migrants and residents by external environmental factors both in the larval and adult stage [28,39], similarly to *Loxostege sticticalis* (Lepidoptera: Pyralidae) [48]. *Mythimna separata* larvae that experienced a high larval density, short photoperiod, high rearing temperature, poor nutrition, and other environmental factors tend to develop into migrants that have a longer POP and stronger flight capacity than residents [47,55,56]. In crickets and planthoppers, environmental condition changes also affected the wing form (long or short/absent) during development [57]. In many migratory insect species, relatively long POP was an important behavioral characteristic of migrants [23,27,48], which might reduce the number of days available for reproduction but result in an increased flight potential and time window to engage in migration [9], such as *L**. sticticalis* (L.) [58], *M. separata* (Walker) [59], and *C. medinalis* (Guenée) [60]. We therefore speculated that rearing temperatures in the immature stage might determine the migration and differentiation of *C. medinalis* adults. Females that experienced low rearing temperatures in the immature stage (18 and 22 °C) had a greater resident propensity with a strong reproductive ability. By contrast, those that experienced high rearing temperatures in the immature stage increased their migratory propensity and motivation [16]. Field studies also indicated that the primary driver of *C. medinalis’* northward migration over the summer months was typically assumed to escape from high temperatures [35].

Temperature can directly influence flight like other insect behaviors by physiological regulations [61,62,63]. Temperatures below or above the optimal threshold temperature negatively affect the flight performance of migratory insects [64], such as *M. separata* [16], *Mythimna loreyi* (Walker) (Lepidoptera: Noctuidae) [49] and *Bactrocera dorsalis* (Hendel) (Diptera: Tephritidae) [62]. Similar results were obtained in our study with the treatments below or above 26 °C in the immature stage showing a relative decreased tendency of flight duration and flight distance. The reason for the negative effect may be closely related with the activity of the metabolic rate and flight muscle power output regulated by temperature [2]. Below the optimal temperature, an insect reduces the locomotor activities of flight muscles, which should be provided energy during flight as a consequence of the lower metabolic rate [65]. High temperatures cause inactivation of the nervous system due to the increase of heat production by flight muscles above the upper critical thermal limit [66].

Interestingly, female moths have a stronger flight propensity than males in some species [67], which undertake the task of finding food and oviposition sites [68]. Guo et al. [45] reported that female *C. medinalis* had a greater migratory propensity and flight capability than males at constant 26 °C in a laboratory population. Similar patterns were observed in *M. separata* [55] and *Culex pipiens pallens* (Diptera: *Culicidae*) [69]. However, differential flight performances between the sexes in response to rearing temperatures in the immature stage are less common in migratory insects. In this study, we further found that flight performances of *C. medinalis* females and males largely depended on rearing temperatures in the immature stage. Females exhibited better flight performance than males at above 18 °C, with greater flight duration, distance, and velocity. By contrast, 18 °C caused poorer flight performance of females than males.

Generally, migration is an energy-intensive activity, along with worse environments, new enemies, and increased risk of mortality [23,26]. Migration-mediated reproductive costs are not absolutely negative, and differ from species to species and the flight stage in some migratory insects [46,70]. Previous studies have demonstrated that an appropriate take-off time or migration stage can weaken the oogenesis-flight syndrome and therefore pay no reproductive costs for their migration in insects [29,46]. Zhang et al. [28] reported that flight on the first day after emergence significantly promoted the reproduction of *C. medinalis* under laboratory conditions. Our study further revealed that the effects of flight on reproduction of *C. medinalis* on the first day after emergence were also mediated by rearing temperatures in the immature stage. Compared to control females, flying females after experiencing low temperature treatments in the immature (18 and 22 °C) started laying eggs at a later age and had lower total fecundity and mating frequency. These results suggested that low immature rearing temperatures negatively affected the migration at the expense of huge reproductive costs. However, the high rearing temperature of 34 °C significantly decreased the POP and increased the synchrony of egg-laying of flying females with a significant decrease in PFO, causing an increase of subsequent larval densities and even population outbreaks. A similar effect has been observed in *L**. sticticalis* [48]. Combined with the decreased reproductive performance and enhanced migratory propensity of *C. medinalis* treated with 34 °C, we therefore supposed that the high temperature treatment (34 °C) in the immature stage was most likely to trigger the onset of migration of *C. medinalis* adults for survival, which was consistent with the main motivation of insect migration triggered as a response to external environmental factors [26]. Supporting evidence for this comes from one previous study, which reported that high larval rearing temperatures increased the activities of metabolic enzymes, restrained the accumulation of energy substances, and benefited migration in *M. separata*, while low larval temperatures promoted the accumulation of energy substances, decreased the activities of metabolic enzymes, and suppressed migration [71]. In China, as a temperate insect, the rice leaf roller cannot overwinter in the area north of 30° N. Only a few individuals can survive in areas of Yunnan, Guangxi, and other coastal areas of southeast China during winter [35,38,72]. Annual migration from the southern overwintering areas to crop production regions in northern areas further expands population distributions and escape from the high temperatures of spring and summer in south China [34]. As a migratory species, the rice leaf roller that encounters temperature variations during the immature stage causes facultative migration, which may help explain the phenomenon of migrant and resident polytheisms observed in other seasonal migrants [9,16,37,49,56], and predict population infestations and outbreaks of *C. medinalis* [60,73].

In insects, fluctuating temperatures and extreme temperatures play extremely important roles in development, longevity, and other life-history characteristics of reproduction [1,74,75,76]. McCalla et al. [74] reported that parasitoids reared under fluctuating profiles at low average temperatures developed faster (15 °C) and survived longer (15–20 °C) when compared to those reared under constant regimes with corresponding means. By contrast, high average fluctuating temperatures produced parasitoids with an extended developmental period (35 °C) and reduced longevity (30–35 °C). Milosavljević et al. [76] also found that the development rate of insects was significantly higher at intermediate temperatures than at cline margins, including low temperatures. Our results were obtained from a set of constant temperature regimes in the laboratory, ranging from 18 to 34 °C, but the specific effect of fluctuating temperatures and extreme temperatures remained unclear for *C. medinalis*. Therefore, studies across a broader set of fluctuating temperature regimes in the future are still necessary to understand the real effect of temperature on the characteristics of the insect pests, as this is the closest to the daily temperature fluctuations that occur in the field. Furthermore, further research on the specific effects of extreme temperatures (lower than 18 °C and higher than 34 °C) on development, adult reproduction, and flight behavior are also needed to extrapolate the applicability of our speculation, which are more critical for the survival and development of this species across different climate scenarios or ranges. Furthermore, our studies were only carried out in lab studies. The current results cannot be directly extrapolated to the field investigation and fully represent the field situation, because the field environment is more complex. It still needs more field evidence results in the future.

## 5. Conclusions

Taken together, our data indicate that the development, reproduction, flight, and migration behavior of *C. medinalis* varied among different rearing temperatures in the immature stage. When the rearing temperature in the immature stage surpasses the optimum for reproduction, *C. medinalis* accelerates maturation development and this facilitates adult migration. However, at larval rearing temperatures below the optimum for reproduction, *C. medinalis* delays maturation development and this weakens the motivation to migrate but accelerates reproduction processes and adult reproduction. These findings enhance our understanding of diverse adaptive strategies in response to different temperatures in insects and still have value in guiding agricultural production, crop protection, occurrence prediction of pests, and integrated pest control to some extent. Further studies are required to test the effects of rearing temperatures in the immature stage on the reproduction and migration of *C. medinalis* moths across a range of fluctuating temperatures from 11 to 36 °C and combined with field trials.

## Figures and Tables

**Figure 1 insects-12-01083-f001:**
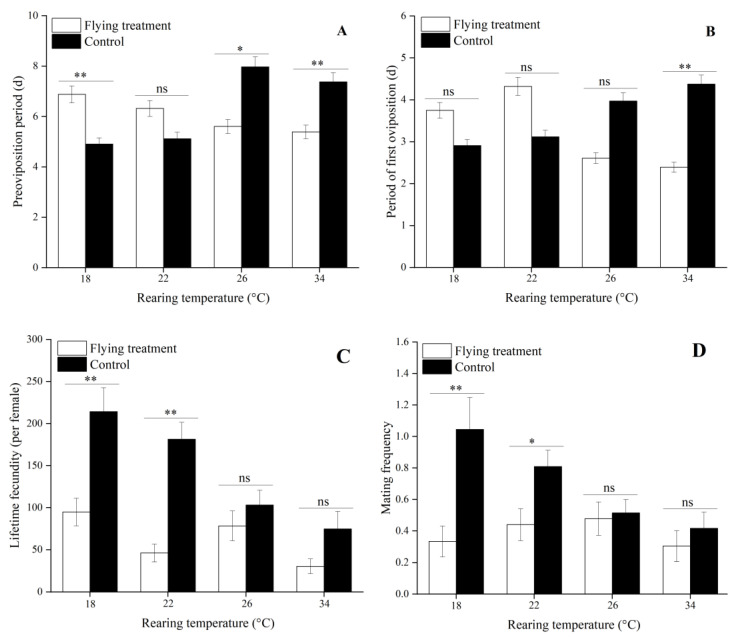
Preoviposition period (POP) (**A**), period of first oviposition (PFO) (**B**), lifetime fecundity (**C**), and mating frequency (**D**) of *C. medinalis* in the flying vs. control groups under different rearing temperatures in the immature stage. Data are presented as mean ± SE. “*”, “**” on bars, indicate significant or highly significant differences in the same bars by the *t*-test (*p* < 0.05 or *p* < 0.01). “ns” above bars indicates no significant difference between the flying and control treatments at the same temperature (*t*-test, *p* < 0.05 or *p* < 0.01). Sample sizes for each flying treatment are 22, 42, 35, and 24 females, and each control treatment are 24, 25, 23, and 23 females, from low to high temperatures, respectively.

**Figure 2 insects-12-01083-f002:**
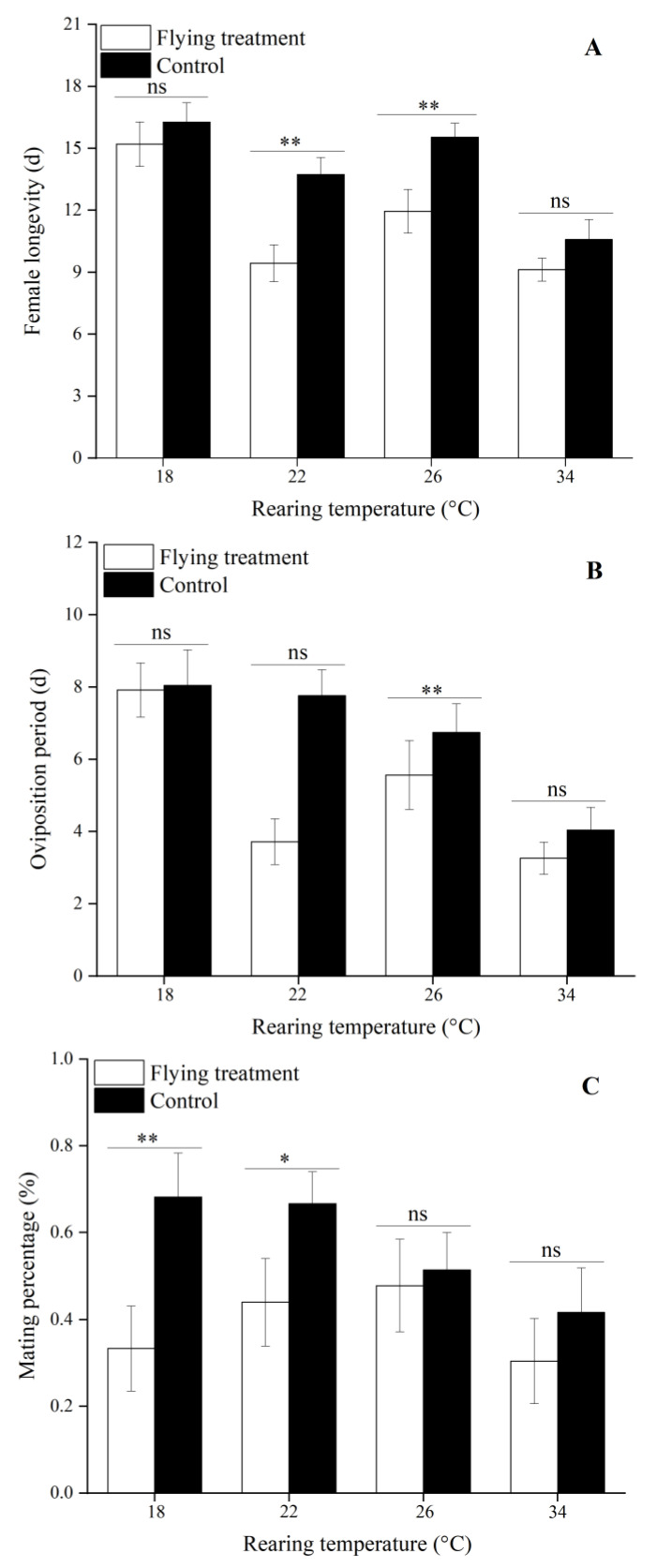
Female longevity (**A**), oviposition period (**B**), and mating percentage (**C**) of *C. medinalis* in the flying vs. control groups under different rearing temperatures in the immature stage. *, ** above bars represent significant or highly significant differences between the flying and control treatments at the same temperature (*t*-test, *p* < 0.05 or *p* < 0.01). “ns” above bars indicates no significant difference between the flying and control treatments at the same temperature (*t*-test, *p* < 0.05 or *p* < 0.01). Sample sizes, from low to high temperatures, in flying treatments are 22, 42, 35, and 24 females, and in control treatments are 24, 25, 23, and 23 females, respectively.

**Table 1 insects-12-01083-t001:** Larval development of *C. medinalis* under different rearing temperatures.

Development Parameters	Rearing Temperatures			
	18 °C	22 °C	26 °C	34 °C
Egg-larval period (d)	53.80 ± 0.39 a	30.59 ± 0.23 b	23.76 ± 0.08 c	23.25 ± 0.14 c
Pupal period (d)	17.83 ± 0.31 a	8.87 ± 0.19 b	5.78 ± 0.09 c	4.99 ± 0.10 d
Larval mortality rate (%)	74.00 ± 10.51 a	49.35 ± 12.52 a	46.00 ± 25.14 a	53.05 ± 18.49 a
Pupal mortality rate (%)	36.45 ± 23.86 a	25.62 ± 8.71 a	18.85 ± 3.45 a	27.70 ± 5.04 a
Pupation rate (%)	35.14 ± 25.47 a	50.65 ±12.52 a	54.00 ± 25.14 a	46.95 ± 18.49 a
Eclosion rate (%)	71.34 ± 8.15 a	59.94 ± 23.97 a	76.64 ± 3.98 a	68.61 ± 6.28 a
Period of synchronized eclosion (d)	9.40 ± 0.48 a	5.66 ± 0.24 b	4.08 ± 0.13 c	4.61 ± 0.17 c
Sex ratio (female/total progeny)	0.55 ± 0.02 a	0.56 ± 0.06 a	0.57 ± 0.04 a	0.52 ± 0.08 a

Means with the different letters in the same row represent significant differences at the 5% level by Tukey’s HSD test. Sample sizes of 18, 22, 26, and 34 °C treatments are 153, 356, 443, and 318, respectively.

**Table 2 insects-12-01083-t002:** Reproductive performance of *C. medinalis* adults (the controls) from different rearing temperatures in the immature stage.

Reproductive Parameters	Rearing Temperatures			
	18 °C	22 °C	26 °C	34 °C
Preoviposition period (d)	4.91 ± 0.42 b	5.12 ± 0.26 b	7.97 ± 0.90 a	7.38 ± 0.48 a
Period of first oviposition (d)	2.91 ± 0.42 a	3.12 ± 0.26 a	3.97 ± 0.90 a	4.38 ± 0.48 a
Lifetime fecundity	214.23 ± 28.31 a	181.40 ± 20.13 a	103.20 ± 17.38 b	74.62 ± 20.83 b
Mating frequency	1.05 ± 0.20 a	0.81 ± 0.10 ab	0.51 ± 0.08 b	0.42 ± 0.10 b
Female longevity (d)	16.27 ± 0.93 a	13.74 ± 0.82 a	15.54 ± 0.68 a	10.58 ± 0.97 b
Oviposition period (d)	8.05 ± 0.98 a	7.76 ± 0.72 a	6.74 ± 0.79 ab	4.04 ± 0.63 b
Mating percentage (%)	68.18 ± 10.16 a	66.67 ± 7.36 a	51.43 ± 8.57 a	41.67 ± 10.28 a

Means with the different lowercase letters in the same row indicate significant differences by Tukey’s HSD test at the 5% level. Sample sizes for each temperature treatment are 22, 42, 35, and 24 females, respectively.

**Table 3 insects-12-01083-t003:** Flight performance of *C. medinalis* males and females from different rearing temperatures in the immature stage.

Parameters	Sex	Rearing Temperatures			
		18 °C	22 °C	26 °C	34 °C
Flight duration (h)	Female	2.34 ± 0.47 a	3.37 ± 0.36 a	3.87 ± 0.37 a	3.25 ± 0.36 a *
	Male	3.09 ± 0.55 ab	3.04 ± 0.34 ab	3.74 ± 0.34 a	2.02 ± 0.34 b
Flight distance (km)	Female	4.42 ± 1.14 a	7.13 ± 1.09 a	8.12 ± 1.13 a	6.30 ± 0.98 a
	Male	6.07 ± 1.29 a	6.34 ± 0.96 a	7.92 ± 0.97 a	4.15 ± 0.91 b
Average velocity	Female	1.54 ± 0.21 a	2.24 ± 0.1 7 a	2.24 ± 0.14 a	2.76 ± 0.18 a
(km/h)	Male	1.75 ± 0.21 a	1.82 ± 0.15 a	1.92 ± 0.13 a	1.67 ± 0.14 a

Different lowercase letters in the same row indicate significant differences by Tukey’s HSD test at 5% level. “*” represents significant differences in the same column by the *t*-test (*p* < 0.05 or *p* < 0.01). Sample sizes, from the low to high rearing temperature treatment are 14, 28, 29, and 26 females, and 14, 30, 28, and 29 males, respectively.

## Data Availability

The data presented in this study are available on request from the corresponding author.

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
