# Peer review of "Flight and Reproduction Variations of Rice Leaf Roller, Cnaphalocrocis medinalis in Response to Different Rearing Temperatures"

_insects, 2021, doi:10.3390/insects12121083_

Round 1
Reviewer 1 Report
Regarding on the current revised manuscript, looks much better than the previous version, but I am still believe the discussion section can be further improved. If the authors can answer these questions and revise them, the manuscript can be published.
- The authorsstated that temperature impacted the development, reproduction and flight capacity of rice leaf roller, and it developed the adaptive strategies to cope with stressful temperatures. It's a very interesting study, but I didn't see in the discussion was a combination of this research and practical application. These are only the results of lab studies, but many studies are carried out indoors and outdoors. It is necessary for the author to explain why field experiments are not carried out. If the author does not conduct a field trial, at least the authors needs to explain that the laboratory study cannot be extrapolated to the field investigation, because the field environment is more complex and the laboratory study cannot fully represent the field situation.
- The authors also need to emphasize the practical implicationsof the study. This insect adopts strategies to cope with pressure and temperature. Why the authors choose pressure and temperature for research and where pressure and temperature are more likely to occur under actual conditions need to be made clear, otherwise the readers will not understand the purpose of this research. I guess the author is trying to explain that this migratory pest is used to deal with different temperature areas and better spread. But I didn't see any concrete practical implications in the abstract or the discussion, it is hard for readers to relate this research to reality, which made the manuscript less attractive.
Author Response
Dear reviewer:
Thanks very much for reviewers' suggestions on our manuscript entitled “Flight and reproduction variations of Rice Leaf Roller, Cnaphalocrocis medinalis in response to different rearing temperatures” (manuscript # insects-1486683). We have read these comments carefully and made major modification correspondingly. We have labelled all the modifications we have made marked up using the “Track Changes” function and also the answers to these questions including line numbers.
Once again, we acknowledge your comments very much, which are valuable and meaningful in improving the quality of our manuscript.
Sincerely yours.
Weixiang Lv
2021-11-28

Reviewer 2 Report
Paper submitted by Ly et al was greatly improved since its original version.
Experimental work is much easier to understand and results sounds for readers interested in the consequences of larval environment to flight and reproduction of adults in a rice pest.
Authors sumarized results as tables and figures instead of text, that is a good point. However, I'm not totally convinced of the usefulness of presenting data in both tables and figures. The editor may ask to choose and include some raw tables in sup files.
Despite a strong attention to spelling and grammar, some additionnal corrections are needed. For instance lines 100, 116, 152, ... and more.
One of my previous suggestion was not fixed. Contrarily to the author's statements, the names of insects in introduction and discussion were not followed by their order name; the name of the discoverer is not useful.
Author Response

(The authors gave the same response as above.)

Reviewer 3 Report
The authors have done a fine job addressing all of my original comments and those of other reviewers. I have no further comments for improvement. Thank you and good luck!
Author Response
Dear reviewer:
Thanks very much for reviewers' suggestions on our manuscript entitled “Flight and reproduction variations of Rice Leaf Roller, Cnaphalocrocis medinalis in response to different rearing temperatures” (manuscript # insects-1486683).
Once again, we acknowledge your comments very much, which are valuable and meaningful in improving the quality of our manuscript.
Sincerely yours.
Weixiang Lv
2021-11-28

This manuscript is a resubmission of an earlier submission. The following is a list of the peer review reports and author responses from that submission.
Round 1
Reviewer 1 Report
The authors did a study on the adaptive strategies of Cnaphalocrocis medinalis at five temperatures, which is a very interesting study. The authors do a lot of work, however, they did not present the results of the study well. I am confused as to whether this adaptive strategy refers to the tradeoff between reproduction and migration or other behavioral characteristics. According to the manuscript, the insects exhibit different adaptive strategies at high and low temperatures. I would suggest that authors rethink their approach to presenting your results in a more straightforward way. I don't think it makes sense to separate flying insects from non-flying insects. It's better to get rid of non-flying insects. In addition, the author considered whether it was necessary to combine male and female insects. Here, gender did not have a great influence, but separate expression led to a very tedious whole article. The author also has too many tables, which makes it difficult to obtain valid information. Some languages also need to be improved, and I suggest finding a native speaker to help with the changes. In general, authors need to do a good job of linking your results to the discussion section and presenting the conclusions more clearly.
Here are the specific questions:
L17 When the Latin name is at the beginning of a sentence, use the full name
L34-36 This statement cannot be inferred from your results above, which do not mention the results of the flight.
L11-36 In simple summary, the author's description of adaptive strategies is confusing, describing the trade-offs between migration and reproduction at high temperatures; at low temperatures, however, there seems to be a trade-off between reproduction and development.
And in the abstract, you don't mention any migration behavior in the results (L28-33), but you mention migration in the conclusion (L34-36), and I don't see any correlation between those two parts.
L100 what is “encountered by C. medinalis in the field” mean? You mean these are the five temperature gradients that they often experience in the field. However, the temperature in the field is dynamic and continuous, and this separated pattern is not possible. What I suggest you do is just say that the temperatures you choose represent your high and low temperatures. Here I would like to emphasize why you chose the temperature of 26℃. I guess this temperature is your control temperature, but you don't mention this key information here.
L167-170 Here you are only comparing the two low temperatures with the other three, but not comparing the two high temperatures with the others, which leads to confusion. You actually studied the effects of both low and high temperatures on this pest.
L170-172 This sentence is supposed to describe the effects of high temperatures, but you only said the effects of 30℃ without stating that this represents high temperatures, leading to confusion.
L172-174 Why don't you say non-significant increase or decrease, and I wish you would separately add the in-text citation of test statistics instead of just putting a P>0.05.
L174-176 Again, I suggest you separate the high and low temperatures for clarity.
L178-181 I recommend that you combine Table 2 and 3, as they are both statements of flight capability.
L192-193 I don't know how you came to that conclusion, this should be in the discussion section, not the results section.
L194 I think you seem to be missing your overall goal here, look at the title your overall goal is to look at the effects of temperature, but here you add the effects of flight. I understand you want to consider flying and non-flying insects in their entirety. But in keeping with the overall goal, I suggest you eliminate non-flying insects and study only the effects of temperature on flying insects.
L194-255 After reading this section, I strongly recommend that you eliminate the effects of flying. It's too complicated to get to the point. As I said above, you're talking about temperature effects all the way through, but here you add the effects of flight, which is confusing.
L319-321 I'm a little confused here. What the author clearly states in “Simple Summary” is the effect of temperature on flight performance: “we investigated the effects of rearing temperatures in the immature stage on immature development, adult reproduction and flight performance of Cnaphalocrocis medinalis, one major migratory pest feeding on rice”. But flight is used as a factor here and in the results section.
L385-388 Again, temperature is the influence factor on flight performance.
L391-414 What you want to express throughout is that the effect of temperature leads to a tradeoff between reproduction and flight, that the reproduction of insects decreases at low temperatures and high temperatures, and that flight increases, leading to energy transfer between reproduction and migration.
I think there is something wrong with your research method. In fact, a better method is to measure the flying ability (male and female) and reproductive ability (female) of all the insects under 5 temperature gradients. Non-flying insects are not included in the analysis. In my opinion, the situation of non-flying is only because there is no flight behavior during the measurement period, which does not mean that it will not fly all its life. Then directly measure the spawning quantity and flight ability at 5 temperatures for comparison.The number of eggs laid is the most important characteristic to measure the insect's ability to lay eggs.
Author Response
Thanks very much for reviewers' suggestions on our manuscript entitled “Adaptive strategies of Rice Leaf Roller, Cnaphalocrocis medinalis in response to different rearing temperatures” (manuscript # insects-1430483). We have read these comments carefully and made major modification correspondingly. We have labelled all the modifications we have made marked up using the “Track Changes” function and also the answers to these questions including line numbers.
Once again, we acknowledge your comments very much, which are valuable and meaningful in improving the quality of our manuscript.
Sincerely yours.
Weixiang Lv
2021-11-12

Reviewer 2 Report
Manuscript submitted by Weixiang et al present original results on the consequences of rearing at different temperatures of the pest moth C medicalis. Data were obtained in lab conditions and discussed at the light of agronomic environments of this migrating insect.
I think biological results interesting and valuable, but presentation must be entirely revised.
Title is "adaptive strategies", this assumption do not correspond to conclusions that could be made from results. "consequences of " or "flight and reproduction variations" would be more accurate.
Key words should be different from title
In many sentences, authors are too finalists when considering the insect behaviour or physiology. For instance line 53, they do not "adjust their physiology to deal with extreme conditions", their physiology change under extreme conditions...
When citing other insects, the name of the finder is not important, but the order must be cited, also in discussion.
In the introduction, only temperature is considered, not other components of the climate. Were for instance moisture or wind considered?
M&Ms.
sentence lines 98-101 is hard to understand
Line 120, sex ratio is always the proportion of males and females, but usually it is calculated as number fo females/total progeny.
Please, indicate the number of individuals when describing an experiment.
line 140, what is a 48 channel flight mill?
Two major parameters are missing, namely body size and survival, I think it is a major concern of this work.
Results
Table 1 do not show the sex of individuals, were parameters different between males and females? Sex ratio has to be calculated as females/total offspring.
Page 5 is very unclear and hard to read because authors repeat stats that are also showed in tables and figures. I suggest to remove all stats in text to allow easy reading.
Table2: results for 30°C are clearly an artifact, they should be intermediate between 26 and 34. Either the authors start over or they delete those results. Also in figures 1 and 2.
Table 5. I don't understand lines "Mating frequency" and "mating rate"; did some individuals mate more than once? If some mate and other not, it should be clearer presented.
Figure 2 results are hard to compare with table 5 data on oviposition period, why presenting some data on tables and others on figure from the same experiment?
Discussion.
lines 310-312. sentence is introduction, begin with a summary of results.
Take care of stats and results. on fig 2, non flying females at 30°C have an oviposition period not statistically different from others, but bars evidence a difference. Stats calculate the risk you take when comparing data, they do not chose the results for you. By changing stats, you may change the conclusions. Please consider this fate in discussion.
Authors consider their tests as "extreme" temperatures. But actually they are not. From -10 to 45°c would be extremes, but not 18 to 34°C.
When reading the discussion, I have the feeling that authors forget that experiments on fligth and reproduction were done at control temperature. So the considerations on muscles and flight are not right. Do they found in the litterature considerations of the acquisition of energy stores during the larval period for a use in the adult?
For flight performance, body size is crucial.
At the end of discussion, do authors consider that this species has two morphs? and that larval temperature induced a switch? This has to be compared with crickets or coleoptera that are able to adapt to stresfull conditions.
Conclusion. No, moths cannot "choose" their adaptive strategy, and no data on survival were given (line 419). This conclusion is disconnected from results. Moreover, pest management needs considerations on parasitoids, survival, diapause, nutritional sources... that were not provided.
If authors aim to conclude on trade offs or reaction norms, they have to follow the template applied in the litterature, and the figures and statistical procedure of linear regressions.
In general, text must be corrected by an english speaker.
Author Response
Thanks very much for reviewers' suggestions on our manuscript entitled “Adaptive strategies of Rice Leaf Roller, Cnaphalocrocis medinalis in response to different rearing temperatures” (manuscript # insects-1430483). We have read these comments carefully and made major modification correspondingly. We have labelled all the modifications we have made marked up using the “Track Changes” function and also the answers to these questions including line numbers. We have deleted the results for 30 °C according to your suggestions.
Once again, we acknowledge your comments very much, which are valuable and meaningful in improving the quality of our manuscript.
Sincerely yours.
Weixiang Lv
2021-11-12

Reviewer 3 Report
The authors have graphed and presented their results clearly, drawing some attention to the implications of their findings. I found the study of interest and a good contribution to the knowledge of bioecology of pest moths. The methods used are appropriate for the objectives of the work and, in general, well depicted. The resulting figures are sufficient, informative, and of good quality helping to follow the reasoning throughout the manuscript.
The Intro and Discussion provide no insight on how this MS relates to the various other ones cited in the text or concerns that have been raised by other researchers. This article should provide details on all these fronts to provide the proper context for the work. Authors do not present any hypotheses or expectations that could be connected to previous studies; adding these details will improve the paper. The authors should clearly explain WHY THE STUDY WAS DONE, WHY IT WAS IMPORTANT, and HOW IT FITS WITH OTHER STUDIES. It should be clear and concise. The intro should also include what outcome(s) they expect, and how it would help support or refute their hypotheses or answer their questions. Some of the authors statement would be much stronger if they tie their work to the body of literature that has built up on fluctuating temperature effects on insect development, longevity, reproduction, and survival (see below).
Next, my main concern is that the authors are extrapolating the applicability of their results beyond what the design supports. The data in this study were developed from five sets of highly artificial constant temperatures between 18C and 34C most of which are closer to the optimal range for development of this pest species (favored temperatures for development of this pest were estimated at 22-29C, see study by Padmavathi et al. 2013, J Insect Sci 13, 96). The development, reproductive, and longevity of C. medinalis at temperatures lower than 18C and higher than 34C, and which are more critical for the survival and development of this species across different climate scenarios or ranges, were not validated in this study. This is a critical limitation of the study, and the authors must concede and discuss this.
More specifically, there are still uncertainties about the results obtained, especially because experiments were conducted in a lab at a set of constant temperature regimes, ranging from 18C to 34C. Therefore, studies across a broader set of fluctuating temperature regimes are still necessary to understand the real effect of temperature on the characteristics of the insect pests, as this is the closest to the daily temperature fluctuations that occur in the field (see studies by McCalla et al. 2019 J. Econ. Entomol. 112: 1560-1574; Milosavljevic et al. 2019 J. Econ. Entomol. 112:1062-1072; Milosavljevic et al. 2020 J. Econ. Entomol. 113: 633-645)). The lower threshold temperatures of 11-12 was estimated by Padmavathi et al. 2013, J Insect Sci 13, 96 for egg, larva, pupa, and total development. The abovementioned work with other pests and natural enemies at fluctuating temperatures has found strong evidence of increased longevity in insects reared at non-stressful low temperatures when compared to higher temperature regimes. This article should provide details on all these fronts to provide the proper context for the work. They also indicated that the consumption rate development of insects was significantly higher at intermediate temperatures than at cline margins, including the low temperatures. Adding these details will improve the paper in my opinion.
This is not to diminish the data gathered in this study, they are of value. But it is important for the authors not to overgeneralize, and to warn the reader, including regulatory agencies, against doing so as well. This article should provide details on all these fronts to provide the proper context for the work.
Overall, I was excited to see the results of the paper after reading the abstract, but I found it hard to extract key messages useful to policymakers and professionals, probably in large part due to the lack of connection with other published work and need for improved structure of the current manuscript.
Specific comments:
L14-15 I’d suggest deleting the part:” … in the immature stage …”; it should read: “… we tested the effects of rearing temperatures on immature development, adult reproduction, …”
L24: Please provide a full scientific name at the first mention of the species in the MS.
L77-79: This study did not investigate how this pest copes with temperature fluctuations (other studies have investigated this for other systems: see my general comments above). The present study investigated the developmental, reproductive and flight capabilities of moths across the range of differing constant temperatures. This should be corrected in my opinion.
L100: Please explain why the range of 18-34C was used in this study. If I can remember correctly, these are the monthly averages typical of March through August in Changde. Monthly averages for Jan-Feb and Sep-Dec are way below 18C. The lower threshold temperatures of 11-12 was estimated by Padmavathi et al. 2013, J Insect Sci 13, 96 for egg, larva, pupa, and total development. Previous work with other pests and natural enemies at fluctuating temperatures (McCalla et al. 2019 J. Econ. Entomol. 112: 1560-1574; Milosavljevic et al. 2019 J. Econ. Entomol. 112:1062-1072; Milosavljevic et al. 2020 J. Econ. Entomol. 113: 633-645) has found strong evidence of increased longevity in insects reared at non-stressful low temperatures when compared to higher temperature regimes. This article should provide details on all these fronts to provide the proper context for the work. They also indicated that the consumption rate development of insects was significantly higher at intermediate temperatures than at cline margins, including the low temperatures. Adding these details will improve the paper in my opinion.
L106-163: When research is performed covering the survival and behavior of a of multistage insect pest, key information on the actual number of insects from each stage and instar surviving the treatment and finally reaching the adult stage is mandatory. Indeed, mortality of each stage/instar should be provided for all treatments (i.e., # of larvae to start with, # of enclosed adults for reproduction and flight performance). There is also no evidence that authors considered sample size or the potential of their experiment to detect an expected difference. Did they conduct a power analysis to establish a sample size that could detect an expected difference? Replication seems very weak in terms of the chances of detecting a difference and across all experiments. With three replicates how much difference could they expect to see?
Author Response

(The authors gave the same response as above.)
